# Influence of Nitrogen Fertilization and Cutting Dynamics on the Yield and Nutritional Composition of White Clover (*Trifolium repens* L.)

**DOI:** 10.3390/plants14172765

**Published:** 2025-09-04

**Authors:** Héctor V. Vásquez, Leandro Valqui, Lamberto Valqui-Valqui, Leidy G. Bobadilla, Manuel Reyna, Cesar Maravi, Nelson Pajares, Miguel A. Altamirano-Tantalean

**Affiliations:** 1Laboratorio de Agrostología, Instituto de Investigación en Ganadería y Biotecnología, Facultad de Ingeniería Zootecnista, Biotecnología, Agronegocios y Ciencia de Datos de la Universidad Nacional Toribio Rodríguez de Mendoza de Amazonas, Chachapoyas 01001, Peru; leandro.valqui@untrm.edu.pe (L.V.); docente1106@untrm.edu.pe (L.V.-V.); leidy.bobadilla@untrm.edu.pe (L.G.B.); manuel.reyna.epg@untrm.edu.pe (M.R.); cesar.maravi@untrm.edu.pe (C.M.); nelson.pajares@untrm.edu.pe (N.P.); 7501770981@untrm.edu.pe (M.A.A.-T.); 2Escuela de Posgrado, Programa Doctoral en Ciencias para el Desarrollo Sustentable, Facultad de Ingeniería Zootecnista, Biotecnología, Agronegocios y Ciencia de Datos de la Universidad Nacional Toribio Rodríguez de Mendoza de Amazonas, Chachapoyas 01001, Peru

**Keywords:** *Trifolium repens* L., performance, adaptability, model, sustainability, biological nitrogen fixation

## Abstract

White clover (*Trifolium repens* L.) is known for its ability to fix nitrogen biologically, its high nutritional value, and its adaptability to livestock systems. However, excessive fertilization with synthetic nitrogen alters its symbiosis with *Rhizobium* and reduces the protein content of the forage. The objective of this study was to evaluate the interaction between nitrogen fertilization (0 and 60 kg N ha^−1^), cutting time, and post-cutting evaluation on the morphology, yield, and nutritional composition of white clover. A completely randomized block experimental design with three factors, distributed in three blocks, was used. Within each block, three replicates of each treatment were assigned (six interactions), giving a total of 54 experimental units. The data were analyzed using a three-way analysis of variance and Tukey’s multiple comparison test. Exponential models and generalized additive models (GAMs) were applied to the morphology and yield data to identify the best fit. The treatment with 60 kg N ha^−1^ and cutting at 30 days showed significant increases in plant height (47.42%), fresh weight (59.61%), dry weight (98.41%), and leaf width (27.55%) compared to the control. It also produced the highest protein content (28.44%) compared to the other treatments with fertilization, without negatively affecting digestibility. The GAMs best fit most morphological and yield parameters (except leaf height and width). All fertilized treatments had higher fresh and dry weight yields. In conclusion, applying 60 kg N ha^−1^ after cutting at 30 days, followed by harvesting between 54 and 60 days, improved both the quality and yield of white clover, which favored sustainable pasture management and reduced excessive nitrogen use.

## 1. Introduction

Clover is a forage legume of great agronomic importance due to its multiple benefits in grazing systems and its suitability for hay and silage production [1]. Among the most important species is white clover (*Trifolium repens* L.), characterized by its wide global distribution, from the Arctic to high and tropical regions, with optimal adaptability in areas with high rainfall [2]. This species provides significant environmental benefits, such as increased earthworm and microorganism activity in the soil, as well as attracting pollinators, which contributes significantly to biodiversity conservation [3,4]. At the local level, livestock farming is one of the main economic activities in more than 50% of the provinces of the Amazonas region, with open-field and silvopastoral livestock systems being the most common [5,6]. In this context, clover becomes an essential forage resource in livestock production, representing up to 70% of the diet of sheep and cattle when available [7,8]. Its high protein content and excellent digestibility promote nutrient intake and assimilation, stimulating livestock growth and development, which translates into higher milk and meat production without reducing forage availability [9,10].

Nitrogen (N) is essential for plant growth, development, and stress response; its deficiency triggers a complex adaptive response characterized by the upregulation of the *VvNPF2.3* gene, which is responsible for the uptake, transport, and efficient use of this nutrient. This activation is associated with a 62.2% reduction in indole-3-acetic acid (IAA) levels and a 21.3% increase in hydrogen peroxide (H_2_O_2_), which alters the plant’s hormonal and oxidative balance, affecting key processes for its development [11]. On the contrary, excessive doses of nitrogen can alter the nitrogen balance in the soil, increasing the risk of nitrate leaching and affecting the mineral nitrogen content [12]. The white clover is distinguished by its ability to fix a significant portion of atmospheric nitrogen through symbiosis with bacteria such as *Rhizobium leguminosarum* sv. *trifolii* in its roots, contributing between 75 and 92% of the total nitrogen absorbed [13,14,15]. This mechanism allows it to thrive in low-nitrogen soils and reduce dependence on synthetic nitrogen fertilizers [15,16]. However, excessive application of these fertilizers can inhibit biological nitrogen fixation, replacing the biological source with a synthetic one [17,18,19], which compromises the nutritional value of the forage, especially by reducing its protein content. However, this effect may be partially offset by an increase in biomass production [20,21,22].

Therefore, the nitrogen dose must be maintained at a level that does not affect the symbiosis; this, combined with optimal cutting timing, becomes a key factor in managing forage yield and quality. Cutting during flowering produces higher green and dry biomass yields, but nutritional quality tends to decrease, while earlier mowing can favor an increase in protein content and improve digestibility, although with lower averages [23,24,25]. In northern Peru, it has been shown that associations between *Trifolium repens* L. and *Lolium multiflorum* L. favor higher biomass and protein content when cutting intervals are strategically managed between 30 and 45 days, which is relevant for livestock feed [26]. Similarly, Salcedo-Mayta et al. [27] report that clover cover can increase quinoa grain yield by up to 17%. This highlights the social and economic importance of clover in the agricultural and livestock sectors.

Plant morphological growth and performance can vary depending on the treatments applied; for this reason, mathematical models are used to make scientific predictions about future changes in plant behavior. These models are based on sequential assessments derived from biological principles, which allow quantitative measurements of plant shape and morphology to be obtained [28]. Among the most flexible models applied in biological and environmental sciences are generalized additive models (GAMs) [29,30,31]. These models offer the advantage of capturing nonlinear and complex relationships between variables, making them suitable for data that do not follow a simple relationship. In contrast, the exponential model, which has been widely used in studies on nutrient uptake rates and corn organ growth [32,33], assumes linear relationships and is useful for describing growth processes that follow a constant rate. The data obtained from these models can be crucial for implementing resource optimization and decision-making strategies, facilitating more efficient and sustainable management of agricultural resources [34,35].

Although it is widely recognized that high nitrogen fertilization in *Trifolium repens* L. can shift the nitrogen source from biological to synthetic, and that the timing of cutting influences the balance between yield and nutritional composition, comprehensive studies are still needed that include evaluations at different cutting times and monitoring of subsequent behavior, incorporating the combined use of mathematical models to predict morphological dynamics and future yields. The main objective of this research was to determine the impact of the interaction between nitrogen fertilization (0 and 60 kg N ha^−1^), cutting time, and post-cutting evaluation time on the morphological parameters, yield, and nutritional composition of white clover (*Trifolium repens* L.). Mathematical models were also applied to predict morphological development and biomass yield. It was hypothesized that the interaction of these factors significantly affects performance parameters, morphology, and nutritional quality, and that at least one of the two mathematical models evaluated would provide a better fit. This study presented alternatives to optimize farmers’ production costs, allowing them to make more informed decisions to improve input efficiency, promoting more profitable and sustainable pasture management by reducing the effect of excessive use of synthetic fertilizers on biological fixation and its environmental impacts.

## 2. Results

### 2.1. Morphological Parameters

The interaction between nitrogen doses, cutting time, and post-cutting evaluation time was significant for plant height and leaf width (*p* < 0.05), while leaf length and stem diameter showed no effects on the interaction (*p* > 0.05), as shown in Figure 1. From day 7 to day 28 after cutting, no significant differences were detected in plant height or leaf width. From day 49 onwards, the application of 60 kg N ha^−1^ resulted in significantly greater plant height compared to the control (without fertilization), with values of 16.67, 18.83, and 19.00 cm at 30, 45, and 60 days of cutting, representing a superiority of 47.42%, 71.18%, and 70.25% regarding control, as shown in Figure 1a; 56 days, values of 23.00, 24.67, and 20.67 cm (86.54%, 102.80%, and 62.03% above the control) were reached. Leaf width responded significantly to nitrogen fertilization in the evaluations at 35 and 56 days after cutting, in treatments where cutting took place at 30 and 45 days, where 40.03 and 39.73 mm were recorded at 35 days, equivalent to increases of 44.20%, 43.12% compared to the control without fertilization with cutting at 60 days; at 56 days, the values were 46.30 and 46.40 mm, exceeding the same control by 27.55% and 27.82%. However, these differences were not statistically superior to the other treatments (Figure 1c). It should be noted that the variables leaf length and stem diameter did not vary over time after cutting, as shown in Figure 1b,d. Overall, the patterns indicate that nitrogen fertilization favors the growth of white clover, with a greater influence on plant height and leaf width (Appendix A).

The behavior of the values observed in the morphological variables over time after cutting can be seen in Table 1. Overall, the exponential models and generalized additive models (GAMs) showed a good fit. The mean bias (MBE) was practically zero, and the coefficients of determination (R^2^) and model efficiency (EF) indices were high, especially for plant height and leaf length, and moderate for leaf width and stem diameter. These patterns indicate that the post-cut dynamics are well explained by the models and that the differences between the approaches respond to the form of the biological response. After cutting, the variables increased continuously and steadily, approaching a maximum value; this trend was also observed for plant height, which showed a sustained increase, except in the treatment with 60 kg N ha^−1^ and cutting at 60 days, where the trend was less uniform. This pattern was better fitted by the exponential model, which describes this type of data very well, showing a stabilizing trend. However, when the response showed more complex variations over time, such as growth acceleration, temporary plateaus, and inflection points in the trend, the GAMs offered better fits in capturing this behavior, except for the 60 kg N ha^−1^ treatment and cutting at 30 days, which showed more constant growth in leaf length, better fitting the exponential model. The results suggested that, under conditions where forage growth is stable, an exponential model is sufficient to estimate post-cut behavior and plan management. However, when responses are more variable over time with sudden changes, the use of GAMs allows for better anticipation of plant development.

Figure 2 shows, for each treatment, the trajectories predicted by the most appropriate model selected based on goodness of fit. The respective equations according to their models can be found in the Appendix A. The exponential models follow clearer patterns, while the GAM captures the nonlinearities, which clarifies how the choice of model varies depending on the nitrogen dose and the cut-off time.

### 2.2. Performance Parameters

A significant interaction was detected between nitrogen dose, cutting time, and post-cutting evaluation time for fresh weight and dry weight yield (*p* < 0.05) (Figure 3a,b). Significant variations were observed between the fertilized treatments and the control, revealing a certain superiority from 28 days after cutting onwards in the fresh weight variable. At 49 days after cutting, the treatment with 60 kg N ha^−1^ and scheduled cutting at 60 days reached 1298.33 g m^−2^, significantly exceeding the control without fertilization with cuts at 30, 45, and 60 days (823.33, 811.67, and 783.30 g m^−2^) by 57.69%, 59.96%, and 63.66%. At 56 days, all nitrogen treatments outperformed the control regardless of the cutting time, with 60 kg N ha^−1^ and cuts at 30, 45, and 60 days (1356.67, 1606.67, and 1446.67 g m^−2^) showing a superiority of 59.61%, 85.38%, and 59.56%, respectively (Figure 3a). In terms of dry weight, 49 days after cutting, the treatments with 60 kg N ha^−1^ and scheduled cuts at 45 and 60 days presented values of 177.03 and 202.53 g m^−2^ of dry biomass, being statistically higher compared to the control with the same cutting times by 32.81% and 54.49%. The treatment with 60 kg N ha^−1^ and cutting at 30 days showed no significant differences compared to the control at the same evaluation point based on dry weight. On day 56, all treatments with nitrogen fertilization and cutting at 30, 45, and 60 days showed the best yields with 224.27, 262.30, and 236.37 g m^−2^ of dry weight, indicating increases of 98.41%, 118.16%, and 88.44% compared to the control (Figure 3b). These findings underscore the positive effect of nitrogen application on biomass production in white (Appendix A).

The behavior of the values observed in the performance variables over time after cutting can be seen in Table 2. In general, the generalized additive models (GAMs) showed a good fit for both variables (fresh weight and dry weight), outperforming the exponential models in all combinations of nitrogen dose and cutting time, with higher model efficiency (EF) and coefficient of determination (R^2^) and lower values of root mean square error (RMSE) and mean bias error (MBE). This superiority indicates a greater ability to capture nonlinearities and interaction effects in post-cut behavior in biomass accumulation. Consequently, GAMs are a viable option for yield prediction and identification of points where yield may vary over the 56 days of evaluation.

Figure 4 shows the behavior of the best-fit models according to the treatment, where the lines represent the values predicted by each model based on the performance parameters. This graphical representation allows for a clearer visualization of the fit of the models. According to the results of the statistical analysis of the performance presented, the equations of the models can be found in the Appendix A.

### 2.3. Bromatological Parameters

A significant interaction was observed between nitrogen dose, cutting time, and post-cutting evaluation time for ash content, acid detergent fiber (ADF), and neutral detergent fiber (NDF) (*p* < 0.05), while moisture did not have significant effects (Figure 5a). In the first post-cut evaluation, the ash content was higher with 60 kg N ha^−1^ and cutting at 30 days (12.52%), exceeding the control with the same cutting time by 16.14% and the treatment with 60 kg N ha^−1^ and cutting at 60 days (10.52%) by 18.98%; in the second and third evaluations, no significant differences were detected (Figure 5b). As for ADF, in the first post-cut assessment, 0 kg N ha^−1^ with cutting at 45 days (26.97%) was higher than 0 kg N ha^−1^ with cutting at 30 days (21.42%) by 20.36%, and by 13.65% and 15.98% compared to the treatment with 60 kg N ha^−1^ with cutting at 45 and 60 days (23.73% and 23.25%). However, no significant differences were observed between the other groups. In the second evaluation, 60 kg N ha^−1^ with cuts at 30 and 45 days (26.17 and 26.08%) exceeded the respective controls (22.59% and 22.79%) by 15.81% and 14.39%, and in the third evaluation, there were no differences (Figure 5c). Regarding NDF, the first post-cutting evaluation showed no differences; in the second, 0 kg N ha^−1^ with cutting at 60 days (30.63%) and 60 kg N ha^−1^ with cutting at 30 days (30.64%) had the highest averages, exceeding 0 kg N ha^−1^ with cutting at 45 days (24.47%), indicating a statistical superiority of 25.16% and 24.82%, respectively, with no differences compared to the other treatments. In the third evaluation, 0 kg N ha^−1^ with cutting at 60 days (30.38%) presented a value higher by 24.77% and 21.85% than the treatments without fertilization with cuts at 30 and 45 days, and 28.72% higher than 60 kg N ha^−1^ with a cut at 30 days (23.60%), with no significant differences from the others (Figure 5d) (Appendix A). 

The results revealed a significant interaction between nitrogen doses, cutting time, and post-cutting evaluation time on protein content, gross energy, and in vitro digestibility of dry matter (DIV) (*p* < 0.05), as shown in Figure 6; however, crude fiber did not show significant differences (Figure 6a). Regarding protein (Figure 6b), in the first post-cut evaluation, the highest value was obtained with 60 kg N ha^−1^ and cutting at 30 days (28.46%), higher than all other treatments, except for unfertilized cuts at 45 and 60 days (28.05 and 26.52%), which showed no differences. In the second evaluation, the unfertilized cuts at 30 and 45 days (25.82% and 26.19%) had the highest protein contents and were statistically superior to their fertilized counterparts at the same cutting times by 13.91% and 32.34%, respectively; both also outperformed all other treatments. In the third assessment, 60 kg N ha^−1^ with cutting at 30 days (28.44%) was superior to the other fertilized treatments, exceeding the 60 kg N ha^−1^ cuts at 45 and 60 days (25.40 and 25.41%) by 43%, but did not differ from the unfertilized controls, regardless of the time of cutting. Regarding gross energy (Figure 6c), the highest value in the first evaluation was observed in the unfertilized cut at 45 days (4817.52 kcal), higher than all treatments except the unfertilized cut at 60 days (4694.55 kcal). No differences were detected in the second evaluation. In the third evaluation, the unfertilized cut at 45 days (4615.65 kcal) exceeded the fertilized cuts at 45 and 60 days (4480.81 and 4470.64 kcal) by more than 7.5%, with no differences from the other treatments. On the other hand, the DIV (Figure 6d) showed the highest values in the first evaluation after cutting, with no differences between treatments, with averages above 92%. In the second evaluation, the unfertilized cuts at 30 and 60 days (80.57% and 80.25%) were statistically higher than the other treatments; in the third evaluation, these same treatments only differed from the 60 kg N ha^−1^ cut at 60 days (Appendix A).

## 3. Discussion

### 3.1. Morphological Variables

The results showed that the interaction between fertilization dose, cutting time, and post-cutting evaluation time significantly affected plant height and leaf width. During the 7 and 28 days of post-cutting evaluations, no significant differences were observed between the treatments. The absence of differences during the first evaluations could be due to the short recovery time after cutting, as previous studies indicate that *Trifolium repens* L. shows more evident recovery at 35 days, when resources are no longer concentrated mainly in roots and shoots [36]. In addition, it has been reported that immediately after cutting, clover reduces the number of branches and leaf size, effects that diminish as recovery progresses [37,38]. In our study, both leaf height and width began to increase from 35 days, and the effect of fertilization became statistically more evident from 49 days after cutting, regardless of the time of cutting. This pattern suggests that a longer recovery period, combined with nitrogen doses (60 kg N ha^−1^), favors the restoration of vegetative structure.

### 3.2. Fresh and Dry Biomass Yield

Yield, evaluated through fresh and dry biomass, underwent significant changes in response to the experimental treatments applied, with and without nitrogen fertilization. Sun et al. [39] pointed out that nitrogen application in legumes can reduce recovery time and increase biomass accumulation. Similarly, in the present study, variations with respect to nitrogen fertilization were significantly noticeable after 28 days, at which point it began to stand out from the treatments without fertilization. In the evaluation at 56 days, all treatments with nitrogen significantly outperformed those without fertilization, possibly due to the immediate availability of synthetic nitrogen, which influences plant growth stimulation [40,41]. However, its rapid volatilization could have limited the utilization of the total nitrogen applied [42,43]. Unlike treatments without nitrogen fertilization, these depended exclusively on nitrogen fixed biologically by *Rhizobium*, a process conditioned by the formation of nodules, the efficiency of the symbiosis, and the energy that the plant allocates to sustain the bacteria, factors that together can significantly influence its development [44,45]. It has been reported that the use of synthetic nitrogen fertilizers can increase biomass by up to 30% more than symbiotic fixation alone [46]. The nitrogen in our study contributed significantly to the increase in yield by more than 50% in fresh weight and more than 80% in dry weight on day 56 after cutting. This variability highlights the importance of efficient nutrient management, especially nitrogen fertilization, in increasing clover yields.

### 3.3. Prediction Models

Generalized additive models (GAMs) showed a better fit in all treatments for yield and in most morphological parameters, with the exception of plant height, where they fit only one of the treatments. This superiority is due to their ability to capture complex nonlinear relationships using smoothed functions (splines), which gives them the flexibility to model growth and yield trajectories that do not follow predefined parametric patterns [47,48]. In contrast, the exponential model assumes a specific functional relationship, making it more appropriate when the behavior of the variable responds to biological processes that follow a constant or proportional rate of change, such as plant height [49,50,51]. The differences in goodness-of-fit values between the models were relatively small, suggesting that both may be suitable for capturing the behavior of growth and yield variables. However, the flexibility of the GAMs indicated moderately higher values, suggesting better capture of data with high or complex variability at certain evaluation points, resulting in more accurate patterns for estimating yields and some morphological variables. This model could be applied to contexts where topography and environmental conditions are similar; otherwise, an error rate of 1 to 58% could occur, hindering prediction accuracy [52,53,54]. Based on these results, the GAMs enabled more accurate predictions to be made, which is crucial in field applications for decision-making based on nitrogen application in white clover, taking into account different cutting times.

### 3.4. Nutritional Composition

The application of nitrogen fertilization had a significant impact, indicating variability in nutritional content, which depended on the time of cutting and the days after cutting. In the first post-cut evaluation, the treatment with 60 kg N ha^−1^ and cutting at 30 days presented an ash content of 12.52%, exceeding the equivalent treatment without fertilization by 16.14%. This trend could be associated with the time of cutting, since early cuts in forages coincide with a phase of rapid growth, where the plant makes the most of nutrient absorption, accumulating a higher concentration of minerals such as phosphorus (P) and potassium (K), which contribute to an increase in ash content [55,56,57]. According to the most recent literature, optimal ash levels in production and domestic animals are between 5% and 10%, indicating that values outside these ranges may affect digestibility [58,59]. Unlike previous studies, our results differ from the ranges presented; however, under our conditions, we can say that ash tends to decrease when maximum growth is reached. Indeed, during our evaluations, we observed that the ash values ranged from 10.52% to 12.52% with fertilization and from 10.78% to 12.74% without fertilization during the first evaluation, while at the end of the third evaluation, the treatments with 60 kg N ha^−1^ cut at 30 days and 0 kg N ha^−1^ cut at 45 days showed significant decreases from the first evaluation to the third after cutting. These results are similar to those of Valqui et al. [60] in *Brachiaria mutica* without fertilization, where they determined an ash content of 7.74% during the first evaluation (30 days), which decreased in the second evaluation (75 days). These results are consistent with our study’s results for the treatments mentioned above. However, they differ from the other treatments, where the ash content did not vary (60 kg N ha^−1^ with cutting at 45 and 60 days, and 0 kg N ha^−1^ with cutting at 30 and 60 days). The results obtained in our research are relevant for identifying the optimal time for using forage, since the decrease in ash content may be associated with the physiological maturity of the plant and changes in the structural composition of the tissue. The variable values obtained with nitrogen applications, in some cases, are not affected, but in others are, depending on the time of cutting, which influences nutritional values and management strategies to maximize quality.

Regarding the acid detergent fiber (ADF) content, during the second post-cut evaluation, the treatments with 60 kg N ha^−1^ and cutting at 30 and 45 days recorded the highest ADF values compared to the control. These results differ from those reported by Karbivska [57], who observed that a dose of 50 kg N ha^−1^ significantly reduced ADF, while 25 kg N ha^−1^ had a lesser effect; he also indicated that a high dose of nitrogen has less effect on neutral detergent fiber (NDF). On the other hand, in our study, the NDF content was lower in the first evaluation when fertilization was applied at the time of cutting at 45 days, compared to the equivalent control. The results presented are consistent with the studies by Zhang et al. [61], who observed decreases in NDF content with high nitrogen applications (240–360 kg N ha^−1^) in intercropped crops. During the third evaluation, cutting at 60 days with fertilization showed a higher NDF content than that recorded with cutting at 30 days under the same conditions. These results could be related to the longer time to cutting, attributed to greater aging, causing regrowth of more lignified tissue; as a result, the NDF content tends to be higher [62,63]. In this sense, there are precedents where this effect could significantly influence the digestibility and protein content of the forage [64,65]. In general, when ADF and NDF contents are high, digestibility is lower [66]. In our study, digestibility did not vary at some evaluation points, where the first evaluation after cutting showed no difference in digestibility between the treatments and remained above 92%. In the second evaluation, the treatments without nitrogen fertilization with cuts at 30 and 45 days showed higher digestibility than their fertilized counterparts; in the third evaluation, the differences were again not significant. It should be noted that, among other variables that can influence digestibility, ADF and NDF are influenced by the intensity and frequency of grazing, as well as the vegetative state of the plant, with younger plants being more digestible and therefore more attractive to animals when selecting food during grazing [67,68].

In this study, crude fiber averages ranged from 15.85% to 17.22% without fertilization and from 18.35% to 19.03% with nitrogen fertilization, with no statistical differences. According to previous studies, a content close to 15.38% is optimal for mature beef cattle, as it promotes an adequate balance of volatile fatty acids and contributes to reducing methane production [69]. On the other hand, Macgregor et al. observed that increasing the crude fiber dose above 13% did not generate advantages and, in fact, concluded that this level covers production requirements without compromising efficiency [70]. However, these results may vary according to the needs of the cattle, breed, and their requirements according to their stage of growth [71]. Based on this criterion, our results indicate that some treatments are outside the recommended range, which could affect both digestibility and forage utilization efficiency. This supports the need to adjust the nitrogen dose and cutting time to stabilize the crude fiber content at near-optimal levels, optimizing nutritional quality and its applications in the cattle diet.

The protein content was higher at the time of cutting at 30 days compared to cuts at 45 and 60 days when nitrogen fertilization was applied. This result contrasts with that reported by Šidlauskaitė and Kadžiulienė [72], who observed that nitrogen application significantly reduced protein content in white clover–perennial ryegrass associations and had the effect of increasing ADF and NDF values. In the context of our study, regulating the fertilization dose to 60 kg N ha^−1^ with cutting at 30 days could mitigate protein loss and, consequently, reduce the impact of nitrogen applications on forage quality. Another factor that could have influenced the results of our study is the stress caused by limited phosphorus availability, a condition that stimulates legumes to absorb nitrogen more quickly and efficiently, favoring its accumulation in aerial tissues [73]. On the other hand, some research indicates that a higher protein content does not always translate into higher digestibility, as this may depend on various factors, such as the species and variety of legume evaluated [74,75].

Our study points to treatment with 60 kg N ha^−1^ and cutting at 30 days as an option that optimizes nitrogen use since, during protein evaluations, it was higher than at other cutting times with fertilization, while energy content did not differ from the highest averages obtained from treatments without nitrogen, except for the first evaluation. This shows a balance in both protein and energy in this treatment. It is important to note that the energy value of forage varies depending on the species, stage of maturity, and technical management applied. In many cases, energy tends to decrease as forage reaches advanced stages of development, so it is recommended to cut at early stages [76,77]. Consequently, low-energy forages do not provide the energy necessary to maximize animal productivity, even if they have adequate protein levels [77,78].

## 4. Materials and Methods

### 4.1. Research Area

The experimental area was set up at the Amazonas Agricultural Experiment Station of the Instituto Nacional de Innovación Agraria (INIA), located at geographical coordinates 6°12′1.58″ S and 77°52′9.37″ W, in the San Juan annex in the province of Chachapoyas, Amazonas region, Peru. The study area is located at an altitude of 2445 m above sea level, with an average annual temperature of 14.05 °C, relative humidity of 78.8%, and annual precipitation of 874 mm (Figure 7).

### 4.2. Experimental Design

The experimental design used was a completely randomized block design (CRBD) with three factors: nitrogen dose (two levels: 0 and 60 kg N ha^−1^), cutting times (three levels: 30, 45, and 60 days), and post-cutting evaluation time. Treatment allocation was performed with independent randomization within each block. For morphology and yield, the evaluation time was assessed at 8 levels: 7, 14, 21, 28, 35, 42, 49, and 56 days after cutting. For bromatology, the evaluation time was considered at 3 levels, namely, 30, 45, and 60 days after cutting, classifying them as first, second, and third evaluations. The experiment was divided into 3 blocks, and within each block, 3 replicates of each treatment were assigned, resulting in a total of 54 experimental units (6 treatments × 3 replicates × 3 blocks).

### 4.3. Installation of the Experimental Area

The land was laid out in experimental units measuring 6 m^2^ (2 m × 3 m) each, with a distance of 0.5 m between plots and 1 m wide access paths around the experimental area. The land was prepared using cultural practices and agricultural machinery (plowing and harrowing), followed by the formation of plots with a shovel and a leveling bar, and then a rake was used to level the area. The irrigation system consisted of sprinklers with a flow rate of 30 L/s, programmed to operate three times a week, which ensured uniform irrigation distribution and prevented possible waterlogging. The design and frequency of irrigation were determined based on the water requirements of the crop and the sowing season, taking into account the physical parameters of the soil, evapotranspiration, and the agroclimatic conditions of the area. Sowing was carried out by broadcasting with a uniform dose of 15 kg ha^−1^.

For weed control, manual weeding was carried out after each cut and during nitrogen application to avoid nutrient competition with weeds. Chicken manure (40 kg ha^−1^) was also applied, which had a nitrogen content of 2.6% in its composition for uniformity cutting, 90 days after sowing, to homogenize the initial conditions. For treatments with chemical nitrogen fertilization (urea), a single dose (60 kg N ha^−1^) was applied. Nitrogen was applied after the three different cutting times (30, 45, and 60 days) established in the experiment in order to subsequently evaluate the effect after cutting.

### 4.4. Evaluation of Indicators

#### 4.4.1. Morphological Parameters

Morphological parameters were evaluated every 7 days until day 56 after each cut. For plant height, five plants were randomly selected per experimental unit and measured from the base of the stem to the apex of the highest leaf, following the methodology of Vásquez et al. [79], using a graduated ruler. Stem diameter was determined according to Cabello and López [80], measuring halfway up the plant height with a digital caliper.

Leaf length and width were measured according to the methodology of Villegas et al. [81], from the ligule to the apex for length, while width was recorded at the midpoint of the leaf length. A graduated ruler was used to measure length, while a digital caliper was used to measure width. Evaluations were performed on five fully developed leaves selected at random per experimental unit.

#### 4.4.2. Performance Parameters

To determine the fresh weight and dry weight, the methodology of Vásquez et al. [79] was followed. The forage was cut manually from a 1 m^2^ area of each experimental unit, leaving a residue to promote regrowth. The cut biomass was weighed using a precision scale; then, to obtain the dry weight, representative subsamples of the fresh biomass were taken and placed in paper bags. The samples were then dried in a drying chamber at a temperature of 65 °C until a constant weight was reached. The fresh weight and dry weight values were expressed in g m^−2^.

#### 4.4.3. Nutritional Composition Parameters

Samples for bromatological analysis were collected from 1 m^2^ of each experimental unit and then transported to the Laboratorio de Nutrición Animal y Bromatología (LABNUT) of the Universidad Nacional Toribio Rodríguez de Mendoza de Amazonas. A total of 200 g of forage was weighed and placed in an oven at 60 °C for 48 h.

A mini fixed-speed mill was then used at 1730 rpm, and a series of protocols were followed to determine crude fiber, energy, protein, ash, digestibility, acid detergent fiber (ADF), and neutral detergent fiber (NDF). Crude protein was measured using the AOAC No. 928.08 method [82], which included digestion of the sample with sulfuric acid and conversion of nitrogen to ammonia, quantified by titration with hydrochloric acid. The ANKOM A200 procedure [83] was used to determine ADF and NDF. ADF was measured with an acid detergent solution, followed by filtration and oven drying, while NDF was evaluated with a neutral detergent solution, alpha-amylase, and sodium sulfite, followed by rinsing with acetone and oven drying.

In vitro digestibility was evaluated using the ANKOM Daisy system, where samples were incubated with a prepared inoculum and buffer solution at 39 °C. Digestibility was calculated based on post-incubation weight according to AOAC [82]. In addition, the moisture content was determined using the AOAC No. 930.15 method [82], and gross energy was evaluated by calorimetry, following the protocol recommended by the manufacturer for the Daisy II incubation system used in the determination of in vitro digestibility [83]. The ash content was determined using the AOAC gravimetric method No. 928.08 [82], using a muffle furnace (Thermo Fisher Scientific, Waltham, MA, USA). Samples weighing 2 g were placed into porcelain crucibles, which were incinerated at 550 °C for 5 h until a constant weight was obtained.

### 4.5. Predictive Models in Morphology and Yield

The generalized additive model (GAM) and exponential model were chosen in order to capture linear and nonlinear patterns, since the data collected in the field showed irregular deviations depending on the days after cutting, requiring approaches capable of simultaneously representing the general shape of the trajectory and parameters with biological interpretation.

To analyze the behavior of morphological and yield parameters, the averages of each treatment were analyzed using two mathematical models (exponential model and GAM) in order to identify the best fit to the observed data. For each model, several performance criteria were evaluated, such as the coefficient of determination (R^2^), model efficiency (EF), mean square error (ERMS), and mean bias error (MBE) [84,85,86]. These criteria allowed for a comprehensive comparison of the predictive power and accuracy of the models used.
$$R^{2}=\frac{\sum_{i=1}^{N}\left(Y_{obs,i}-{\bar{Y}}_{obs})(Y_{pre,i}-{\bar{Y}}_{pre}\right)}{\sqrt{\sum_{i=1}^{N}{\left(Y_{obs,i}-{\bar{Y}}_{obs}\right)}^{2}\sum_{i=1}^{N}{\left(Y_{pre,i}-{\bar{Y}}_{pre}\right)}^{2}}}$$
$$EF=\frac{\sum_{i=1}^{N}{\left(Y_{obs,i}-Y_{i,\mathrm{obs}mean}\right)}^{2}-{\sum_{i=1}^{N}\left(Y_{pre,i}-Y_{exp,i}\right)}^{2}}{{\sum_{i=1}^{N}\left({MR}_{exp,i}-{MR}_{exp,mean}\right)}^{2}}$$
$$ERMS=\frac{1}{N}\sum_{i=1}^{N}{{\left((Y_{obs,i}-Y_{pre,i}\right)}^{2})}^{\frac{1}{2}}$$
$$MBE=\frac{1}{N}\sum_{i=1}^{N}\left(Y_{obs,i}-Y_{pre,i}\right)$$
where $Y_{obs,i}$ and $Y_{pre,i}$ represent the observed and predicted values, respectively, of the evaluated trait for the *i*-th observation. ${\bar{Y}}_{obs}$ and ${\bar{Y}}_{pre}$ correspond to the mean values of the observed and predicted data, respectively. N denotes the total number of observations considered.

#### 4.5.1. Generalized Additive Models

A generalized additive model (GAM) was used for each treatment combination defined by the interaction between the nitrogen fertilization dose and the cutting time, and the evolution of morphological and yield variables as a function of the days after cutting. The continuous predictor of the model was the time elapsed since cutting (7, 14, 21, 28, 35, 42, 49, and 56 days), while the response variable corresponded to the average of each trait within each treatment combination. This approach used smoothing functions to model nonlinear associations, which allowed flexible patterns in the behavior of the predictor variables to be captured without assuming a specific parametric form [87,88]. The general structure of the model is expressed as:
$$Y_{i}=\beta_{0}+f_{1}\left(t_{i}\right)+\epsilon_{i}$$
where $Y_{i}$ is the observed value of the response variable (plant height, leaf length, leaf width, stem diameter, fresh weight, or dry weight) for the i-th observation, $\beta_{0}$ is the intercept, and $f_{i}\left(t_{i}\right)$ it is a smoothed function of the predictor $t_{i}$ (days after the cut-off: 7, 14, 21, 28, 35, 42, 49, and 56 days), and $\epsilon_{i}$ is the random error. The smoothing function was adjusted using penalized regression splines, with the number of base functions (knots) set at 6. This approach allowed us to model nonlinear relationships between the days after cutting and each response variable, capturing specific patterns for each treatment defined by the interaction between fertilization dose and cutting time.

#### 4.5.2. Exponential Model

An exponential model was applied to describe the relationship between time and the dependent variable, with the aim of predicting adjusted values based on morphological traits (plant height, leaf length, leaf width, stem diameter) and biomass yield (fresh and dry weight) [89]. In this context, $Y_{i}$ represents the observed value of the evaluated trait for the i-th observation, and $t_{i}$ is the number of days after cutting (7, 14, 21, 28, 35, 42, 49 and 56 days). The model is expressed as:$$Y_{i}=a\times e^{{b\;t}_{i}}$$

In this formulation, *a* is the scale parameter, which represents the estimated initial value of the trait at the time of sampling (*t* = 0), and *b* is the rate of change parameter, which can reflect an increase (*b* > 0) or a decrease (*b* < 0) over time. The term *e* corresponds to the base of the natural logarithm.

### 4.6. Statistical Analysis

The evaluations were carried out under a completely randomized block design (CRBD). The workflow consisted of verifying the assumptions of normality (Shapiro–Wilk) and homogeneity of variances (Bartlett) for each variable; the data that met both assumptions were analyzed using three-way ANOVA (fertilization, cutting time, and days after cutting) with a significance level of 5%, followed by Tukey’s test, and those that did not meet normality were log-transformed. Statistical analysis was performed in RStudio (version 4.5.0) using the base package stats [90] and agricolae [91], and graphical visualizations were produced with ggplot2 [92]. Predictive model analysis was performed in Python (version 3.12.4) using the LinearGAM function from the pyGAM package [93] to fit generalized additive models with smoothing terms based on penalized cubic splines, and the SciPy [94] and NumPy [95] libraries to calculate the exponential model, considering time (days after cutting) as the main predictor and each morphological and yield indicator as response variables, with the aim of capturing nonlinear patterns and estimating parameters with biological interpretation for each treatment combination.

## 5. Conclusions

Treatment with 60 kg N ha^−1^ and cutting at 30 days showed the best balance between forage yield and nutritional composition, exceeding the control by 59.61% in fresh biomass, 98.41% in dry weight, 47.42% in plant height, and 27.55% in leaf width on day 56 after cutting. In addition, it recorded the highest protein content in the third evaluation (60 days) and maintained digestibility comparable to that of treatments without fertilization, with lower neutral detergent fiber content than cutting at 60 days without nitrogen.

These results show that combining a moderate dose of nitrogen with a harvest range of 54 to 60 days can optimize the productivity–quality ratio, promoting more efficient use of resources and contributing to the sustainable intensification of forage production. Although biological nitrogen fixation was not quantified, the strategy could be integrated into plans to reduce excessive use of synthetic fertilizers without compromising yield.

It is recommended to validate this combination of dose and cutting time under different soil and climate conditions, simultaneously evaluating productive, nutritional, economic, and environmental indicators, in order to broaden its applicability and contribute to more resilient and efficient systems in the face of the challenges of livestock production.

## Figures and Tables

**Figure 1 plants-14-02765-f001:**
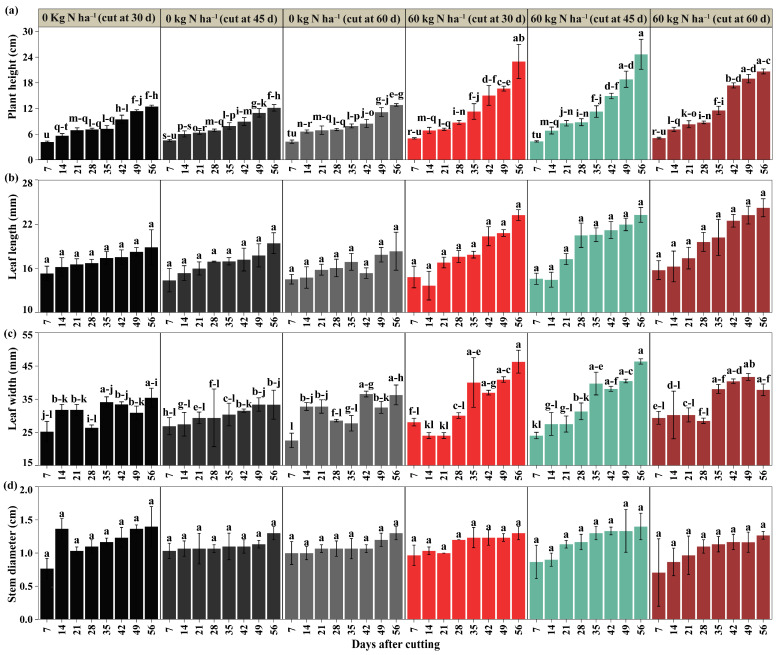
Morphological evaluation of plant height (**a**), leaf length (**b**), leaf width (**c**), and stem diameter (**d**). Significant differences are indicated by different letters vertically according to Tukey’s test. Letter range (-).

**Figure 2 plants-14-02765-f002:**
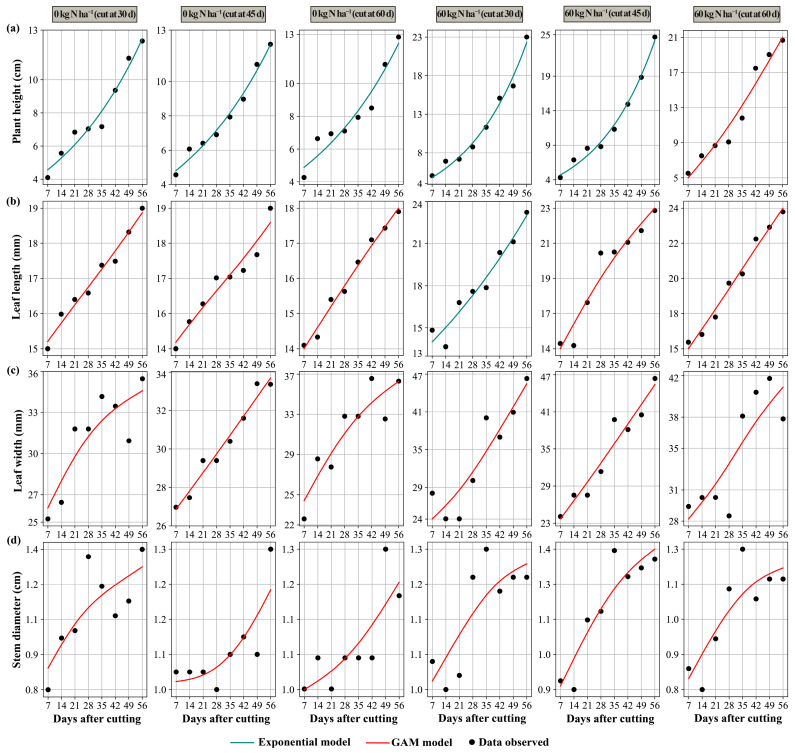
Modeling of morphological parameters of plant height (**a**), leaf length (**b**), leaf width (**c**), and stem diameter (**d**).

**Figure 3 plants-14-02765-f003:**
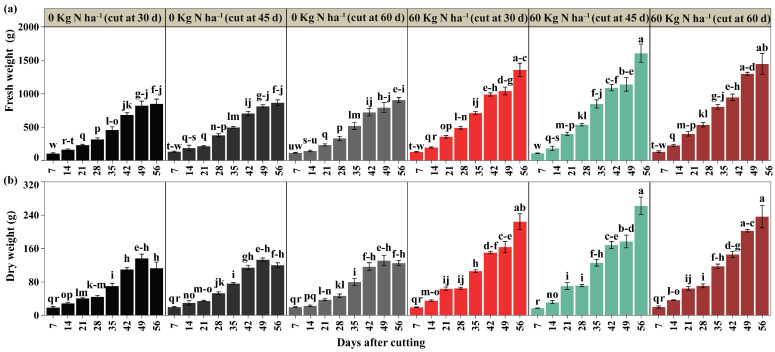
Performance evaluation of fresh forage (**a**) and dry forage (**b**). Significant differences are indicated by different letters vertically according to Tukey’s test. Letter range (-).

**Figure 4 plants-14-02765-f004:**
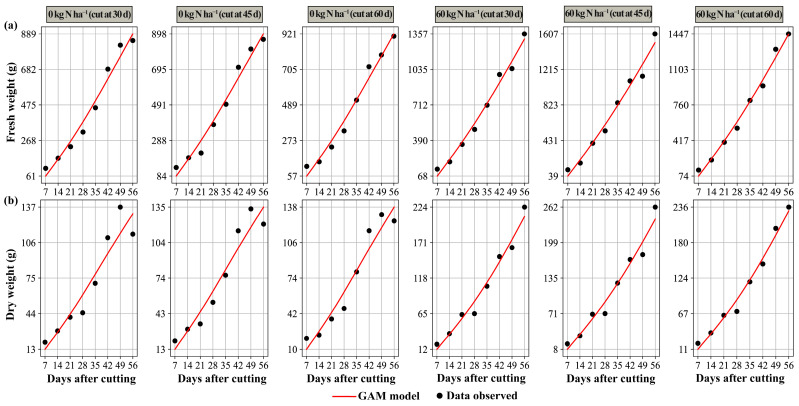
Modeling of performance parameters in fresh weight (**a**) and dry weight (**b**).

**Figure 5 plants-14-02765-f005:**
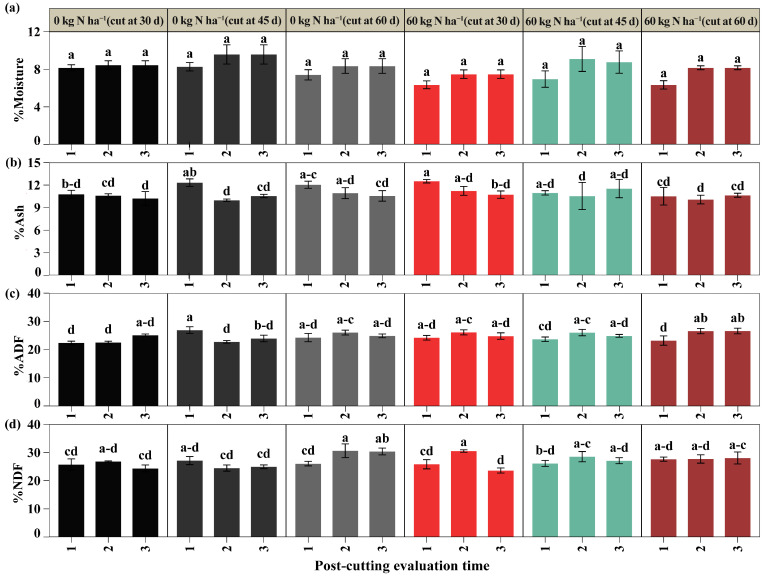
Bromatological evaluation of % moisture (**a**), % ash (**b**), % acid detergent fiber (ADF) (**c**), and % neutral detergent fiber (NDF) (**d**). Significant differences are indicated by different letters vertically according to Tukey’s test. Letter range (-).

**Figure 6 plants-14-02765-f006:**
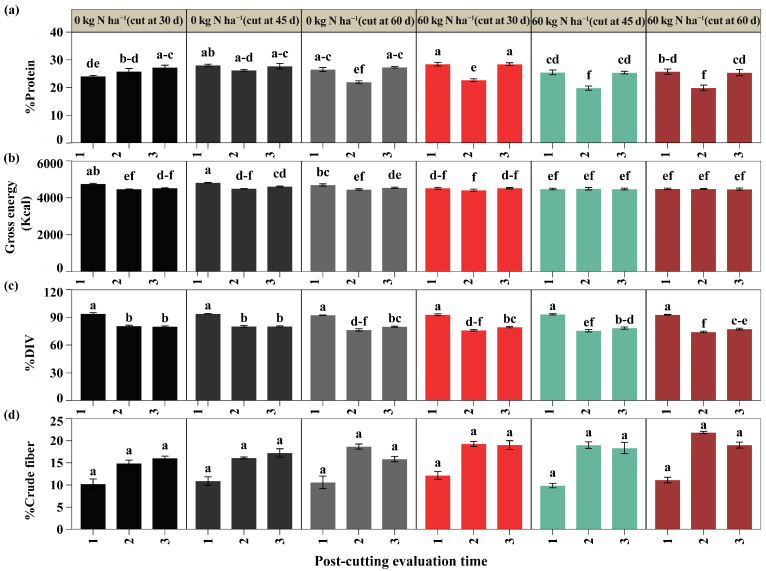
Bromatological evaluation of % protein (**a**), % crude energy (**b**), % in vitro dry matter digestibility (DIV) (**c**), and crude fiber (**d**). Significant differences are indicated by different letters vertically according to Tukey’s test. Letter range (-).

**Figure 7 plants-14-02765-f007:**
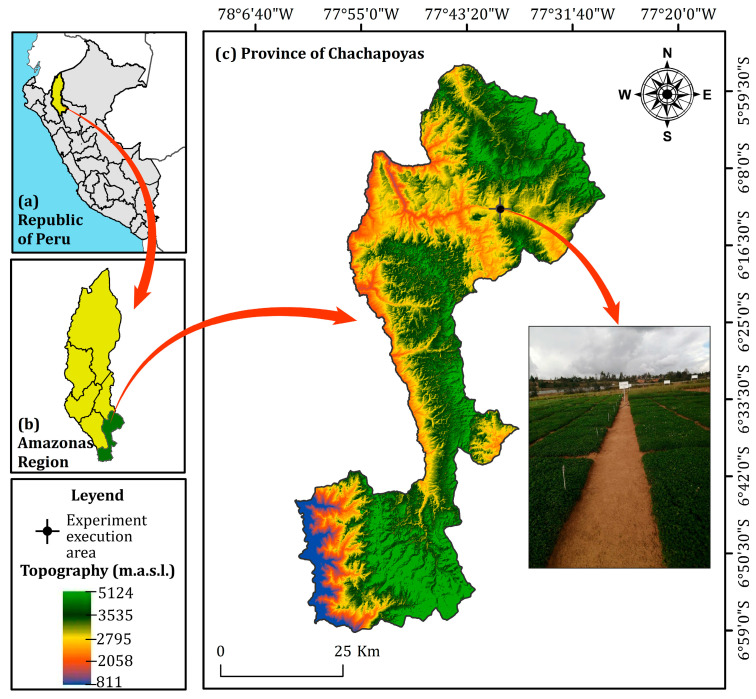
Georeferencing of the experimental area.

**Table 1 plants-14-02765-t001:** Evaluation of mathematical models for white clover morphology.

Parameters	Treatments		GAM				Exponential Model			
	Dose (kg N ha^−1^)	Cutting Time (Days)	MBE	RMSE	R^2^	EF	MBE	RMSE	R^2^	EF
Plant height (cm)	0	30	0	0.5266	0.9799	0.9601	**0.0001**	**0.5021**	**0.9817**	**0.9638**
		45	0	0.3616	0.9887	0.9775	**0.0034**	**0.3129**	**0.9916**	**0.9832**
		60	0	0.6547	0.9660	0.9330	**0.0077**	**0.6395**	**0.9676**	**0.9361**
	60	30	0	1.0346	0.9836	0.9671	**0.0305**	**0.6890**	**0.9928**	**0.9854**
		45	0	1.1101	0.9848	0.9695	**0.0337**	**0.6546**	**0.9947**	**0.9894**
		60	**0**	**1.1307**	**0.9792**	**0.9588**	−0.0361	1.1953	0.9768	0.9539
Leaf length (mm)	0	30	**0**	**0.1630**	**0.9883**	**0.9767**	0	0.1680	0.9876	0.9753
		45	**0**	**0.3542**	**0.9691**	**0.9392**	0	0.3723	0.9658	0.9328
		60	**0**	**0.1568**	**0.9929**	**0.9859**	−0.0003	0.1890	0.9897	0.9795
	60	30	0	0.7126	0.9718	0.9443	**0.0014**	**0.6987**	**0.9729**	**0.9465**
		45	**0**	**0.8049**	**0.9667**	**0.9344**	−0.0073	1.0589	0.9417	0.8865
		60	**0**	**0.3906**	**0.9916**	**0.9833**	−0.0019	0.4424	0.9893	0.9786
Leaf width (mm)	0	30	0	**1.6090**	**0.8802**	**0.7730**	−0.004	1.9983	0.8063	0.6499
		45	0	**0.4107**	**0.9836**	**0.9675**	−0.0001	0.4138	0.9833	0.9669
		60	0	**1.7426**	**0.9183**	**0.8426**	−0.0092	2.1970	0.8661	0.7499
	60	30	0	**3.0548**	**0.9213**	**0.8486**	0.0111	3.1219	0.9176	0.8419
		45	0	**1.8365**	**0.9684**	**0.9378**	−0.0076	1.8817	0.9668	0.9347
		60	0	**2.6924**	**0.8502**	**0.7226**	−0.0029	2.8373	0.8318	0.6919
Stem diameter (cm)	0	30	0	**0.1142**	**0.8088**	**0.6522**	−0.0004	0.1313	0.7353	0.5404
		45	0	**0.0416**	**0.8489**	**0.7131**	0	0.0525	0.7369	0.5429
		60	0	**0.0556**	**0.8172**	**0.6673**	0	0.0590	0.7904	0.6248
	60	30	0	**0.0693**	**0.8176**	**0.6673**	−0.0001	0.0779	0.7617	0.5801
		45	0	**0.0718**	**0.9292**	**0.8627**	−0.0006	0.0947	0.8726	0.7609
		60	0	**0.0780**	**0.8619**	**0.7408**	−0.0003	0.0960	0.7798	0.6078

Note: In each combination of nitrogen dose and cutting time, the values in bold indicate the best fit of the model.

**Table 2 plants-14-02765-t002:** Evaluation of mathematical models for white clover yield.

Parameters	Treatments		GAM				Exponential Model			
	Dose (kg N ha^−1^)	Cutting Time (Days)	MBE	RMSE	R^2^	EF	MBE	RMSE	R^2^	EF
fresh weight (g m^−2^)	0	30	**0**	**46.4716**	**0.986**	**0.9721**	−6.9601	68.0347	0.971	0.9403
		45	**0**	**43.2283**	**0.9874**	**0.9749**	−6.1257	65.604	0.9719	0.9423
		60	**0**	**41.9226**	**0.9894**	**0.9790**	−7.4969	68.1581	0.9732	0.9444
	60	30	**0**	**50.7474**	**0.9924**	**0.9848**	−9.5011	80.3657	0.9819	0.9620
		45	**0**	**71.5865**	**0.9892**	**0.9785**	−12.1286	102.1064	0.9791	0.9563
		60	**0**	**46.1199**	**0.9948**	**0.9897**	−10.6269	82.6629	0.9843	0.9668
dry weight (g m^−2^)	0	30	**0**	**13.1246**	**0.9483**	**0.8992**	−1.0854	16.1611	0.9223	0.8472
		45	**0**	**10.9155**	**0.9657**	**0.9325**	−1.1370	14.7676	0.9381	0.8764
		60	**0**	**10.6767**	**0.9698**	**0.9405**	−1.2808	14.8332	0.9429	0.8852
	60	30	**0**	**10.3831**	**0.9877**	**0.9755**	−1.0195	10.6328	0.9875	0.9743
		45	**0**	**14.4083**	**0.9829**	**0.9660**	−1.3948	14.9280	0.9822	0.9636
		60	**0**	**9.2666**	**0.9920**	**0.9840**	−1.3514	10.7129	0.9899	0.9786

Note: In each combination of nitrogen dose and cutting time, the values in bold indicate the best fit of the model.

## Data Availability

The original contributions presented in the study are included in this article; further inquiries can be directed to the corresponding author.

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
