# Peer review of "Influence of Nitrogen Fertilization and Cutting Dynamics on the Yield and Nutritional Composition of White Clover (Trifolium repens L.)"

_plants, 2025, doi:10.3390/plants14172765_

Round 1

Reviewer 1 Report

Comments and Suggestions for Authors

Vásquez et al. have selected an interesting and relevant topic, focusing on nitrogen (N₂) fertilization in white clover. The manuscript is generally well-presented and informative. However, the following points should be considered to further improve the quality of the paper:

  1. The term "bromatology" may be unfamiliar to many young researchers. It is recommended to either briefly elaborate on this term in the abstract or introduction or consider removing it from the title to improve accessibility and clarity.
  2. Line 98; Beginning the paragraph with “Figure 1” is not ideal stylistically. The authors should first explain the rationale for evaluating these specific parameters before referring to the figure. A brief contextual statement would improve the flow and understanding.
  3. The introduction would benefit from a more detailed explanation of the importance of N fertilization in legume crops like white clover, along with a clearer discussion of the consequences of N deficiency. After establishing this background, the authors can more effectively highlight the research gap. The following reference may provide useful insights and context: http://dx.doi.org/10.3390/horticulturae11030252
  4. The manuscript contains lots of typo’s mistake. Authors need to check the carefully

Reviewer 2 Report

Comments and Suggestions for Authors

This manuscript is based on a field experiment that systematically evaluates the effects

of nitrogen application rate, cutting time, and sampling time on the morphology, biomass,

and nutritional quality of white clover. The study incorporates GAM and exponential models

to analyze growth dynamics. The research has practical significance, with a relatively sound

experimental design, and provides useful insights for white clover management in

high-altitude regions, demonstrating a degree of applied innovation.

However, the study primarily focuses on validating known patterns and optimizing

parameters under specific scenarios, lacking mechanistic breakthroughs and theoretical

depth. The manuscript also contains numerous issues related to language usage, model logic,

and structural organization. These include misused terminology (e.g., “moisture ratio”),

inconsistent variable symbols, and redundant content between the results and discussion

sections, all of which significantly undermine the scientific rigor and readability of the paper.

In summary, I recommend a major revision (Major Revision). The manuscript may

only be reconsidered if the authors can systematically reconstruct the methodological

framework, standardize model descriptions, and improve the overall clarity of language and

structure.

Major Revision Comments:

  1. Abstract: Language and logical flow need further refinement.

(1) It is recommended to unify the verb tense throughout the manuscript, preferably using the

past tense.

(2) The phrase “represents a sustainable strategy” should be revised to “may represent a

sustainable management approach,” unless supported by explicit evidence of ecological or

environmental benefits.

(3) Units should be standardized (e.g., “kg N ha⁻¹”) and used consistently throughout the

manuscript.

(4) The conclusion should clearly emphasize the practical implications of the findings for

grassland management.

  1. Introduction: The section should be streamlined and refocused on the synergistic effects

of nitrogen fertilization × cutting time on white clover yield and nutritional quality.

(1) The logical flow is unclear, especially during the transition from “ecological benefits” to

“nutritional regulation.” The discussion lacks a central focus on how nitrogen application and

cutting time interact to affect yield and quality. Paragraphs should be reorganized to strengthen

causal linkages and better articulate the research motivation.

(2) While the manuscript mentions a lack of integrative studies, it does not clearly state the

limitations of previous research (e.g., single-factor focus on nitrogen, lack of dynamic

multi-time-point assessments, or neglect of biological nitrogen fixation interference). The

scientific problem and its practical relevance should be more explicitly defined.

(3) Core references should be selected carefully to support key arguments. Avoid clustered or

excessive citations such as [5–9], [13–18], which reduce the impact of the evidence.

  1. Results:

(1) A consistent structure is recommended for each subsection: statistical significance →

indicator trend → treatment comparison.

(2) The repeated use of generic expressions like “Figure X shows that...” adds limited value.

Consider replacing with more interpretive and comparative statements such as: “As shown in

Figure 1a, plant height increased significantly under nitrogen fertilization, particularly after day

49.”

(3) In Lines 97–237, in addition to percentage changes, the actual observed values of each

variable should be provided to enhance clarity and comparability.

(4) The application of GAM and exponential models is innovative. However, the current

presentation (mainly via tables and brief summaries) lacks meaningful interpretation or emphasis

on modeling insights.

(5) In Section 2.3, descriptions of nutritional indices (e.g., protein content, ADF, NDF) are

fragmented. Consider summarizing overall trends and highlighting significant differences between

treatments.

(6) Each subsection currently ends with data reporting. Add a short concluding sentence

summarizing the main trend and optimal treatment.

  1. Discussion:

(1) The current structure is a single, lengthy paragraph. It is advisable to break it into sections

by variable type (e.g., morphology, biomass, nutritional traits, modeling).

(2) Language is verbose and occasionally repetitive. Simplify sentence structures and

eliminate redundant connectors (e.g., “...however...nevertheless...”).

(3) There is a weak alignment among results, interpretations, and citations. Focus more on

connecting mechanistic explanations to empirical data with appropriate literature support. For

instance, ADF and NDF patterns are described in detail, but underlying mechanisms are

insufficiently explored, and supporting references are not well matched.

(4) The discussion of model performance notes that GAM outperformed the exponential

model but stops at fit comparison. Deeper insight is needed into the theoretical suitability and

practical application of each model, including why certain variables are better captured by one

model over the other.

  1. Materials and Methods:

(1) The sentence “a single level was applied, beginning after the scheduled cuts at 30, 45, and

60 days” is ambiguous—does fertilization occur before or after cutting? Please clarify the

timeline.

(2) The subsection titled “2.4 Evaluation of indicators” is misnumbered and should be

corrected to “4.4.”

(3) Descriptions of leaf length and width measurements are inconsistent. Although the

method of Villegas et al. is cited for both, only leaf width is explained. Furthermore, two

contradictory methods for leaf width (ligule to tip vs. mid-blade) are mentioned. Each variable

should be clearly defined, and measurement standards unified. For the Weende method, a simple

citation is insufficient—please provide a brief description of key steps (e.g., acid/base digestion,

ash correction) to improve reproducibility.

(4) The term “moisture ratio” (e.g., “MRexp,i,” “MRpre,i”) appears in the modeling section

but is unrelated to the traits measured in this study and is not defined or assessed. These references

appear to be remnants from a different manuscript and should be entirely removed.

(5) The notation used in the GAM and exponential models is inconsistent. Variables are

poorly defined, subscripts are irregular, and time variables are ambiguously expressed. It is

important to clearly define all symbols, identify dependent and independent variables (e.g.,

biomass, plant height), and explain parameter meanings to ensure clarity.

(6) The rationale for choosing each model (GAM vs. exponential) for different variables is

not provided. The authors should clarify that GAM is more suited to nonlinear traits (e.g., leaf

width, dry weight) while exponential models may better fit variables showing consistent growth

(e.g., plant height). A table summarizing the variable–model pairings and justification would

strengthen methodological rigor.

(7) The statistical analysis section, while broadly appropriate, appears somewhat

disorganized. It is recommended to restructure it into four components: workflow, software, data

structure, and hypothesis testing.

  1. Conclusion:

(1) Avoid repeating detailed data from the results section. Focus on summarizing the key

findings.

(2) Be cautious with ecological extrapolations such as the “balance between nitrogen fixation

and chemical fertilization,” especially when nitrogen fixation was not directly measured.

(3) Emphasize the productivity–quality trade-off advantage of the 60 kg N ha⁻¹ + 30-day

cutting treatment and its potential for sustainable management.

  1. Figures and Tables:

The overall layout is clear, but figure resolution and text readability can be further improved. 

Reviewer 3 Report

Comments and Suggestions for Authors

Line 1–2: The context is well established, but quantifying the global impact of drought stress on forage crops would enhance the relevance.

Line 2-3: Clarify what is meant by "cutting dynamics"—e.g., cutting frequency or timing?

Line 5: Ensure the author list and affiliations are consistent with journal formatting.

Line 13–16: Well framed. The risks of excessive nitrogen fertilization are relevant; citing recent studies here would add authority.

Line 17–20: Good explanation of variables studied. Consider simplifying phrasing for broader accessibility.

Line 21–22: Clearly describes the design. Indicating the total number of replicates per treatment would improve clarity.

Line 23–26: Results are effectively summarized. Suggest clarifying what “optimal balance” refers to—nutritional yield, sustainability, or both?

Line 27–29: The conclusion is strong. Emphasize how this informs sustainable agronomy practices.

Line 31: Good keyword selection. Consider adding “biological nitrogen fixation” as a keyword.

Line 33–41: Good overview of white clover's ecological and agronomic value. Suggest including a figure or schematic summarizing benefits.

Line 42–48: Strong rationale for its use in livestock systems. Consider citing more region-specific data.

Line 49–57: Discusses biological nitrogen fixation well. Clarify which Rhizobium strains are relevant.

Line 58–67: Provides strong justification for cutting time as a variable. Could benefit from more critical comparison of findings across cutting intervals.

Line 68–78: Use of modeling is well introduced. Briefly contrast exponential vs. GAM models in practical terms.

Line 79–92: The research gap is well articulated. Suggest explicitly stating the hypotheses tested.

Lines 97–113: Statistical trends are well explained. Visual representation is helpful, but figures would benefit from clearer axis labels and legends.

Table 1: Comprehensive, but dense. Suggest bolding best-performing model values for quicker interpretation.

Line 131–134: Ensure figures are referenced in logical sequence. Confirm all supplementary materials are cross-validated.

Lines 138–154: Interpretation is strong. Clarify whether differences are statistically significant across all timepoints.

Table 2: Like Table 1, useful but visually crowded. Group treatments visually or use shading.

Line 171–174: Add error bars in figures for clarity and statistical strength.

Lines 178–204: Interpretation is mostly clear. Specify if differences in ash or fiber content are practically significant for livestock nutrition.

Lines 209–237: Good identification of trends in protein and digestibility. Suggest discussing implications for ruminant performance explicitly.

Figures 5 & 6: Improve figure legends by specifying treatment groups clearly.

Lines 242–265: Effective link to existing literature. Avoid redundancy by condensing where results and literature are restated too similarly.

Lines 266–275: Modeling discussion is solid. Further emphasize how these results can be generalized or adapted in similar agro-ecological zones.

Lines 276–291: Good interpretation of ash and ADF values. Could be strengthened with comparison to industry forage quality standards.

Lines 292–319: Balanced discussion. Clarify if fiber fractions observed are within optimal livestock ranges.

Lines 320–334: Energy content and digestibility are discussed well. Add a final synthesis paragraph here to reiterate practical takeaways for farmers or agronomists.

Lines 336–344: Geographic details are comprehensive. Suggest including a map scale and orientation in Figure 7.

Lines 345–353: Design is sound. Clarify if randomization was performed independently within each block.

Lines 354–369: Procedures are thorough. Consider stating actual nutrient content of chicken manure.

Lines 371–387: Measurements are standard and well cited. Indicate which plant development stage was targeted for morphological traits.

Lines 389–409: Analytical protocols are appropriate. Ensure consistency in temperature units (°C vs C) and drying durations.

Lines 410–450: Model equations are described in appropriate detail. Brief explanation of why GAM was selected over other nonparametric options would be beneficial.

Lines 451–467: Software and statistical tools are clearly listed. Consider including code or scripts in supplementary materials for transparency.

Lines 468–488: Strong summary. Emphasize broader implications (e.g., contribution to sustainable intensification or climate resilience).

Line 480: Be specific about what “optimal” means—economic, ecological, or nutritional?

Line 487: Recommending further studies under different environmental conditions would strengthen the conclusion.

Formatting: Ensure all tables and figures conform to journal style. Superscripts for units (e.g., kg N ha⁻¹) should be consistent.

Language: The manuscript is mostly fluent but could benefit from light English editing for smoother phrasing in some sections.

References: Comprehensive and relevant. Check formatting uniformity (e.g., DOI placement, spacing).

Round 2

Reviewer 2 Report

Comments and Suggestions for Authors

That's great. I have no other comments on the revisions.

Reviewer 3 Report

Comments and Suggestions for Authors

Done.